# A Debittered Complex of Glucose-Phenylalanine Amadori Rearrangement Products with β-Cyclodextrin: Structure, Molecular Docking and Thermal Degradation Kinetic Study

**DOI:** 10.3390/foods11091309

**Published:** 2022-04-29

**Authors:** Xiaotong Wu, Baoshang Fu, Yunjiao Ma, Liang Dong, Ming Du, Xiuping Dong, Xianbing Xu

**Affiliations:** National Engineering Research Center of Seafood, School of Food Science and Technology, Dalian Polytechnic University, Dalian 116034, China; wxt98321@163.com (X.W.); fubaoshang@dlpu.edu.cn (B.F.); mayunjiao1028@163.com (Y.M.); dongliang@dlpu.edu.cn (L.D.); duming121@163.com (M.D.); dxiuping@163.com (X.D.)

**Keywords:** Amadori rearrangement products, β-cyclodextrin, complex, debittering, flavor

## Abstract

Non-volatile flavor precursors could be used to overcome the flavor loss problems of volatile flavor enhancers during long-term storage. Glu- and Phe-derived Amadori rearrangement products (ARPs) produce pleasant aroma tones thermally but are bitter. We used β-cyclodextrin (β-CD) for debittering Glu-Phe ARPs. ITC analysis indicated that CD-ARP complexes with 1:1 stoichiometry were obtained. NMR analysis indicated that the aromatic ring of Glu-Phe ARPs was embedded in the β-CD cavity. Molecular docking simulations of the bitter taste receptor hT2R1 showed that CD-ARP complex was inactive compared to Glu-Phe ARPs. Complexation with β-CD resulted in the thermal stabilization of Glu-Phe ARPs and a decrease in the degradation rate constant. Compared to Glu-Phe ARPs, the CD-ARP complex in the thermally treated food system slowed down the formation of browning compounds but didn’t inhibit flavor compound formation. The CD-ARP complex is a promising flavor enhancer for applications in flavored and heated foods.

## 1. Introduction

The Maillard reaction (MR), one of the most important reactions in thermal processing, can contribute to the formation of flavor and color and simultaneously affect the nutritional properties of thermally treated food [1]. A recent study demonstrated that the addition of nonvolatile MR flavor precursors, such as Amadori rearrangement products (ARPs), to induce the formation of aroma components may provide an effective strategy to solve the problem of volatile flavor loss during long-term storage [2]. It is expected that the addition of ARPs can induce the generation of volatile aroma compounds prior to eating through a series of rearrangement and degradation reactions during the subsequent thermal processing of food (baking, cooking, microwave heating, etc.) [3]. Therefore, ARPs as novel food additives that promote the formation of aroma compounds could be applied as promising flavor enhancers.

A series of ARPs are usually synthesized using fusion [4], syrup [5] and reflux [6] methods. Recently, new auxiliary methods have been applied to the synthesis of ARPs. For example, a natural deep eutectic solvent (NADES) system with low water content [7], thermally controlled reactions combined with continuous vacuum dehydration [8], spray dehydration [9], and freeze drying [10] have been employed. These methods can be applied to the synthesis of ARPs that generate characteristic flavors. In particular, Glu-Phe ARPs formed from L-phenylalanine (Phe) and glucose (Glu) could produce a pleasant violet flower aroma, which was interesting for flavor enhancement [11]. However, our previous taste test indicated that the addition of Glu-Phe ARPs could induce a bitter taste in foods. This problem has not been adequately investigated. An effective and simple method for debittering Glu-Phe ARPs is required to promote the application of Glu-Phe ARPs in food systems. Commonly, β-cyclodextrin (β-CD) is used to encapsulate bitter molecules as a simple debittering method to reduce the bitter taste of food for a long time [12]. However, whether or not β-CD can be applied as a simple method for debittering Glu-Phe ARPs remains unclear. Similarly, whether β-CD encapsulation of Glu-Phe ARPs affects the degradation rate of Glu-Phe ARPs and the subsequent food quality, such as aroma or color qualities, is also unclear.

In this study, we systemically investigated the debittering mechanism of β-CD-encapsulated Glu-Phe ARPs using nuclear magnetic resonance (NMR), isothermal titration calorimetry (ITC), and molecular docking calculations. The degradation kinetics of Glu-Phe ARPs affected by β-CD encapsulation were also studied. Finally, a complex of Glu-Phe ARPs with β-CD (CD-ARP complex) was used to prepare baked cookies. The aroma and color qualities of the treated cookie products were evaluated to verify the application of the CD-ARP complex.

## 2. Materials and Methods

### 2.1. Chemicals

D-glucose (Glu, 99.5%), L-phenylalanine (Phe, 99%), β-cyclodextrin (β-CD, 98%), sodium bisulfite (NaHSO_3_, AR), and formic acid (chromatographic grade) were purchased from Macklin (Shanghai, China). Ethanol, glycerol, and acetic acid were purchased from Damao Chemical Reagent Co., Ltd. (Tianjin, China). Methyl alcohol (chromatographic grade) was purchased from Merck (99.9%, Darmstadt, Germany). Ultrapure water (UPW) was used for all the experiments.

### 2.2. Synthesis of Glu-Phe ARPs, CD-ARP Complex and Physical Mixture (ARPs + CD)

Glu-Phe ARPs were prepared as previously described in the literature with slight modifications [6]. Glu (0.2 mol), NaHSO_3_ (2 g), ethanol (60 mL) and glycerol (30 mL) were placed in a round-bottom flask and heated in a water bath. When the temperature of the solution reached 80 °C, Phe (0.05 mol) and acetic acid (8 mL) were added. The mixture solution was heated at 80 °C for 5 h. Then, the sample was immediately cooled to room temperature using iced water.

Semi-preparative reversed-phase high performance liquid chromatography (RP-HPLC) with a C18 column (50 μm, 26.8 × 381 mm) was used for the purification of Glu-Phe ARPs. The elution rate was set at 20 mL min^−1^ using a linear gradient from 5% to 50% methyl alcohol over 50 min. The purified Glu-Phe samples were retained and freeze-dried. Mass spectrometry (MS) data were obtained using an ion trap mass spectrometer (Thermo LXQ, Waltham, MA, USA) equipped with an electrospray ionization (ESI) source. The ESI-MS settings were as follows: spray voltage: 4.5 kV; capillary voltage, 37 V; sheath gas, 40 arbitrary units (AU); auxiliary gas, 10 AU. MS spectra were recorded across mass-to-charge ratios (*m*/*z*) in the range of 50–1000 in the positive mode. MS (*m*/*z*): 328.10 [M + H]^+^. The fragmentation patterns of the purified samples were consistent with those reported previously (Appendix A) [13]. NMR data β-pyranose form: ^1^H NMR (400 MHz, DMSO-d6) δ 7.307–7.212 (ddd, J = 10.3, 7.7, 5.0 Hz, 5H, PhH), 3.792 (m, 1H, H-n), 3.543 (m, 1H, H-m), 3.536 (d, J = 5.5 Hz, 2H, H-l), 3.744 (m, 1H, H-o), 3.693–3.550 (m, 1H, H-h), 3.246–3.203 (m, 2H, H-j), 2.973–2.910 (m, 2H, H-g); ^13^C NMR (101 MHz, DMSO-d6) δ 172.24 (C-i), 137.94 (C-d), 129.21 (C-c + C-e), 128.32 (C-b + C-f), 128.15 (C-a),102.68 (C-k), 77.25 (C-n), 76.35 (C-o), 69.71(C-m), 63.27 (C-l), 61.04 (C-h), 52.00 (C-j), 37.22 (C-g).

The CD-ARP complex was prepared using the freeze-drying method. Glu-Phe ARPs were dispersed in 100 mL of β-CD aqueous solution at a molecular ratio of 1:1 and mixed by stirring at room temperature for 6 h. The processed solution was lyophilized for 48 h to yield a solid CD-ARP complex.

The physical mixture (ARPs + CD) of Glu-Phe ARPs and β-CD at a 1:1 molar ratio was also prepared for comparison with the CD-ARP complex. Solid Glu-Phe ARPs and β-CD were ground manually in a mortar for 5 min to obtain a homogeneous blend.

### 2.3. Thermal Degradation Kinetics of Glu-Phe ARPs and CD-ARP Complex

A total of 0.1 g of Glu-Phe ARPs or CD-ARP complex was heated in an oven at 80 °C, 90 °C, and 100 °C, respectively. Samples were heated for different times (0.5, 1, 2, 3, 4 and 5 h) and extracted with 5 mL methyl alcohol. The obtained supernatant was used for the HPLC-diode array detector (DAD) analysis. The obtained data were fitted using a first-order kinetic model (Equation (1)), where C_0_ is the initial Glu-Phe ARPs content, and C is the Glu-Phe ARPs content after a predetermined time (t). The degradation rate constants (k) were derived from the slope of the natural logarithmic curve of the Glu-Phe ARPs retention (ln[C/C_0_] versus time [t]). For the first-order reaction, the half-life values (t_1/2_, the time needed for 50% degradation of Glu-Phe ARPs) were determined using Equation (2).
(1)$$\ln\left(\frac{C}{\mathrm{Co}}\right)=-\mathrm{kt}$$
(2)$$t_{1/2}=-\ln\left(0.5\right)/k\;$$

### 2.4. Preparation of Cookies

The cookies were made of 17 g of low-gluten flour, 6 g of egg liquid, 5 g of corn oil, and 4 g of sugar. Moreover, equivalent ARPs (0.1, 0.2, 0.3, 0.4 and 0.5 g) of Glu-Phe ARPs, CD-ARP complexes or mixtures of Glu and Phe (Glu + Phe) which could theoretically generate Glu-Phe ARPs (0.1 g and 0.5 g) were added to the cookies, respectively. The cookies were baked in an oven at 170 °C for 13 min.

### 2.5. HPLC-DAD Analysis of Glu-Phe ARPs

Glu-Phe ARPs were analyzed using HPLC (Agilent Technologies Inc., Santa Clara, CA, USA) with an SPD-M20 A DAD system. All samples were cleaned using C18 solid-phase extraction (SPE; Agela Technologies, Tianjin, China) before injection. Samples (10 μL) were injected into an XBridge Shield RP C18 column (4.6 × 250 mm, 5 μm) and separated at 25 °C. The mobile phases were 0.1% formic acid in water (solvent A) and 0.1% formic acid in methyl alcohol (solvent B) at a flow rate of 1 mL/min with the following gradient: 0–10 min, 2% B; 10–15 min, 2–100% B; 15–25 min, 100% B, and then back to the initial conditions in 0 min. All samples were detected using a DAD at a wavelength of 275 nm. A single peak of purified Glu-Phe ARPs was observed at a retention time of 7.9 min (Appendix A).

### 2.6. ITC

The interactions between Glu-Phe ARPs and β-CD were investigated using ITC (TA Instruments Co., New Castle, DE, USA). ITC was performed by the titration of 44 mmol L^−1^ Glu-Phe ARPs aqueous solution into 300 μL of 4.4 mmol L^−1^ β-CD aqueous solution at an injection volume of 2 μL and injecting 25 times. The titration parameter reaction temperature and stirring speed were set at 25 °C and 500 rpm, respectively. To correct the thermal effects of mixing and diluting, a control experiment was carried out by injecting the aqueous Glu-Phe ARPs solution into water. The stoichiometric ratio (n), binding constant (Ka), enthalpy change (ΔH), entropy change (ΔS), and Gibbs free energy change (ΔG) were obtained by analyzing the titration data using the NanoAnalyze software.

### 2.7. Fourier Transform Infrared (FT-IR) Spectroscopy

FT-IR (Perkin Elmer, Waltham, MA, USA) was used to measure the changes in the positions and intensities of the infrared absorption peaks corresponding to the functional groups of the samples. After accurately weighing 100 mg of dried spectroscopic-grade potassium bromide (KBr) and 1 mg of the tested samples, the mixture was ground into a fine powder, loaded into a mold, and pressed into tablets. FT-IR spectra were collected in the wavelength range of 400–4000 cm^−1^.

### 2.8. X-ray Diffraction (XRD)

Crystallinity was confirmed by XRD using a Shimadzu XRD-7000S diffractometer with Cu Kα radiation (50 kV, 200 mA, λ = 0.154 nm). All samples were measured in the 2θ angle range of 10–70° and a step size of 0.02°. The degrees of crystallinity were estimated from the XRD pattern simulation using the MDI Jade 6.5 software.

### 2.9. Thermogravimetry-Differential Scanning Calorimetry (TG-DSC)

A Q2000 TG-DSC synchronous thermal analysis instrument (TA Instruments Co., New Castle, DE, USA) was used to determine the thermal stability and thermal characteristics of the samples. The samples (4 mg) were heated in an alumina tray at a heating rate of 10 °C /min from 30 °C to 500 °C under nitrogen flow (50 mL/min).

### 2.10. NMR Spectroscopy

^1^H-NMR spectroscopy of the Glu-Phe ARPs, β-CD, CD-ARP complex, ^13^C-NMR spectroscopy of the Glu-Phe ARPs, and 2D Rotating Frame Overhauser Effect Spectroscopy (ROESY) of the CD-ARP complex were performed using a Bruker DRX 400 MHz spectrometer (Bruker Bio Spin, Karlsruhe, Germany), running at 25 °C (298 K). The solid powder was dissolved in 0.5 mL dimethylsulfoxide (DMSO-d6), and then transferred to a 5 mm PABBO probe. For 2D ROESY experiments, the sample was dissolved in 0.55 mL of DMSO-d6/D_2_O (10:1, *v*/*v*) and equilibrated for at least 24 h. 

### 2.11. Color Determination

The colors of different cookie samples, which were crushed and wrapped in a transparent wrap, were measured using a color photometer (Ultra Scan PRO color, Reston, VA, USA). The color value was reported in the CIE-lab scale as *L** (brightness), *a** (redness), and *b** (yellowness). The value of Δ*E* representing the color difference between samples was calculated using Equation (3).
(3)ΔE=ΔL*2+Δa*2+Δb*212

### 2.12. Headspace Solid-Phase Microextraction Combined with Gas Chromatography-MS/MS (HS-SPME-GC-MS/MS) Analysis

The characteristic flavor compounds benzaldehyde and phenylacetaldehyde produced by the food system were analyzed using HS-SPME-GC-MS/MS. All cookie samples were crushed, and 0.5 g was placed into a 10 mL headspace vial, and 10 µL of the internal standard cyclohexanone (50 mg/L) was added prior to collection. Volatile compounds were extracted in headspace vials and incubated for 20 min in a 60 °C water bath. The SPME fiber (divinylbenzene/carboxen/polydimethylsiloxane; 50/30 µm; Sigma Aldrich, St. Louis, MI, USA) was exposed to the headspace for 30 min in a headspace vial containing the samples. The extracted compounds were thermally desorbed by injecting the fiber into the injection port of the GC system at 250 °C for 5 min and operated in splitless mode.

GC-MS/MS (7890B-7010B, Agilent Technologies Inc., Santa Clara, CA, USA) was equipped with a 30 m × 250 µm × 0.25 µm HP-5-ms column. The test heating program was as follows: At 40 °C for 3 min, temperature increased at the rate of 5 °C/min to 80 °C, then to 160 °C at the rate of 10 °C/min, then the temperature was kept at 160 °C for 0.5 min, then increased to 175 °C at the rate of 2 °C/min, then to 230 °C at the rate of 10 °C/min, and finally kept at 230 °C for 7 min. The carrier gas was helium with a flow rate of 1.7 mL/min. The mass data collected in the EI mode at an energy voltage of 70 eV was used for a full scan at *m*/*z* 50–500. The quantitative ion of benzaldehyde was *m*/*z* 106 and of phenylacetaldehyde was *m*/z 91.

### 2.13. Molecular Docking

The 3D structure (Appendix A) of the investigated compounds as docking ligands was constructed using Gaussview6.0. Quantum chemistry calculations were performed employing the Gaussian 16 software to optimize the geometry and frequency calculations at the B3LYP level with the standard 6–311 g (d, p) basis set [14]. The calculated frequencies for all monomolecular compounds were positive, confirming the stability of the optimized geometry. The bitter receptor hT2R1 model was constructed using homologous modelling [15]. The Ramachandran plot of the model suggested that the hT2R1 model is reliable. To choose the docking method and evaluation function, the compounds were docked into hT2R1 using the Discovery Studio/CDOCKER protocol. The active site was obtained from receptor cavities, with coordinates (x: −53.75, y: −17.43, z: 39.24) and radius of 12 Å.

### 2.14. Statistical Analysis

The means and standard deviations of each experiment were calculated. Analysis of variance (ANOVA) was used to determine significant differences (*p* < 0.05) between each experiment using the SPSS software package (IBM SPSS Statistics 20).

## 3. Results and Discussion

### 3.1. Interaction between Glu-Phe ARPs and β-CD

ITC, an important method for explaining binding mechanisms [16], is used to determine the binding constant and thermodynamic parameters of the interaction between Glu-Phe ARPs and β-CD. Figure 1 showed representative ITC curves for Glu-Phe ARPs titrated into the β-CD solution and integrated into an independent-site binding model. The stoichiometry (n) was 1.003, indicating that β-CD contained one Glu-Phe ARPs molecular chelating site and that the formed CD-ARP complex was at 1:1 stoichiometry. ΔH was negative (−0.504 kJ mol^−^^1^), indicating that the binding of Glu-Phe ARPs and β-CD is an exothermic reaction. When the temperature was increased, the binding was easily dissociated. ΔG was negative (−22.47 kJ mol^−1^), indicating that the interaction between Glu-Phe ARPs and β-CD occurred spontaneously. The binding effect was accompanied by a positive entropy change (ΔS = 55.74 J mol^−1^ K^−1^), which explained that the binding of Glu-Phe ARPs led to a more disordered β-CD structure. When β-CD was combined with Glu-Phe ARPs, the solvent molecules in the β-CD cavity shifted into the solution, and the water molecules in the hydrated shell layer were removed to become free water molecules. In addition, the combination was an enthalpy-entropy jointly driven process mainly driven by entropy (|TΔS|>|ΔH|). The ΔH value of the binding between Glu-Phe ARPs and β-CD was very small, indicating that the main binding force was hydrogen bonding [17]. In addition, the hydrophobic interaction is a long-range interaction that is also important in the case of β-CD complex formation. Similar results have been reported in a comparative thermodynamic study of the formation of natural and hydroxypropyl cyclodextrins with benzoic acid complexes [18].

### 3.2. Identification of CD-ARP Complex Formation

The FT-IR spectra of Glu-Phe ARPs, β-CD, CD-ARP complex, and ARPs + CD are shown in Figure 2a. Among them, Glu-Phe ARPs had the typical characteristics of substituted benzene compounds with characteristic peaks at 3367 cm^−1^, 3030 cm^−1^ and 1627 cm^−1^, corresponding to the OH and C-H on the benzene ring, and the C=C stretching vibration peak in the benzene ring conjugated system, respectively [19]. The characteristic peaks of β-CD were 3391 cm^−1^, 2928 cm^−1^, 1644 cm^−1^, 1157 cm^−1^, and 1028 cm^−1^, corresponding to the O-H stretching vibration, C-H stretching vibration, H-O-H bending vibration, C-O stretching vibration, and C-O-C stretching vibration, respectively [20]. The spectra of the CD-ARP complex and ARPs + CD were similar to the superposition of Glu-Phe ARPs and β-CD, with no new peaks observed, suggesting that the binding between the host and guest was an intermolecular interaction. However, it is noteworthy that the peaks of β-CD at 3391 cm^−1^ was shifted to 3368 cm^−1^ after embedding Glu-Phe ARPs, which might be related to the formation of intramolecular hydrogen bonds and changes in the structure of the hydrated bond [21].

Figure 2b presented the XRD patterns of the samples. The Glu-Phe ARPs did not show crystal diffraction peaks, which indicated that Glu-Phe ARPs were in an amorphous state. The peaks of β-CD were strong and sharp, indicating that β-CD had an obvious crystallization property. The ARPs + CDs were almost identical to β-CD. However, in the pattern of the CD-ARP complex, the peaks of β-CD at 2θ = 12.52° weakened, and the diffraction peaks at 2θ of 11.64°, 13.48°, 14.64°, and after 30° disappeared. Compared with β-CD, the crystallinity of the CD-ARP complex was significantly reduced (from 67.14% to 29.67%). This might be due to the replacement of bound water in the β-CD cavities by Glu-Phe ARPs during complex formation, leading to a decrease in the crystal water content and disruption of the β-CD crystals [22].

DSC analysis was performed by comparing the thermal behaviors of the different samples (Figure 2c). The Glu-Phe ARPs began to react when heated, and the endothermic peak at 133 °C corresponded to decomposition and volatilization. β-CD exhibited a broad and strong endothermic peak (ΔH = 274.6921 J/g) at 90 °C, which was generated by the dehydration of crystal water in the cavity [23]. The endothermic dehydration peak of ARPs + CD was essentially unchanged compared to that of β-CD. However, the peak of the CD-ARP complex shifted to a low temperature by 14 °C, and the intensity (ΔH = 139.2644 J/g) significantly decreased. This phenomenon was due to the fact that the crystallinity of the CD-ARP complex was smaller than that of β-CD with a significant reduction in crystalline water in the cavities [24]. Glu-Phe ARPs were encapsulated in the cavity of β-CD, which led to a change in the thermal properties of β-CD.

Figure 2d,e showed the TG and differential thermogravimetry (DTG) curves of the samples. The TG and DTG curves showed that Glu-Phe ARPs decomposed near 135 °C, which is consistent with the DSC results. The continuous weight loss of Glu-Phe ARPs from the initial temperature to 500 °C indicated that the composition of the sample was complex. The thermogravimetric loss of β-CD before 100 °C was attributed to the release of water molecules, and the thermal weight loss at 300–400 °C was attributed to the degradation of β-CD at high temperatures [25]. The ARPs + CD were distinguished from the CD-ARP complex by the joint weight loss of both Glu-Phe ARPs and β-CD. It can be seen from the DTG curve of the CD-ARP complex that the dehydration characteristic peak of β-CD moved forward, and the decomposition characteristic peak of Glu-Phe ARPs moved toward the high temperature. The main reason was that the Glu-Phe ARPs were included in β-CD, which improved the thermal stability of Glu-Phe ARPs. These results confirmed the formation of CD-ARP complexes and their structural characteristics.

### 3.3. Analysis of the Structure and Binding Mode of the CD-ARP Complex

The NMR spectrum was applied to elucidate the possible interaction between Glu-Phe ARPs and β-CD, as it provided strong evidence of spatial proximity between the guest and host molecules. The Glu-Phe ARPs exist mainly in four tautomeric forms because of the conformational instability of the glycosyl moiety, namely, β-pyranose, α-furanose, β-furanose, and α-pyranose, and the stability decreases sequentially according to the analysis of ring tension and substituent site resistance. The number of open-chain forms is small. β-Pyranose (Figure 3a) was chosen for the analysis because of its high content (about 69%) [6].

The encapsulation of Glu-Phe ARPs in the cavity of β-CD was demonstrated by the chemical shift (Δδ) change (upfield or downfield) observed using ^1^H NMR, calculated as follows: Δδ = δ complex − δ free (Table 1). It has been reported that when ΔδH-3 > ΔδH-5, the guest molecule is partially contained in the cavity, whereas when ΔδH-3 ≤ ΔδH-5, complete inclusions occur because H-3 is located near the wider edge of the β-CD cavity and H-5 is located near the narrower edge of the cavity [26]. In the presence of Glu-Phe ARPs, ΔδH-3 > ΔδH-5 was observed, and the protons located inside β-CD (H-3, H-5, and H-6) moved more efficiently compared to those located outside (H-1, H-2, and H-4). This suggested the formation of a CD-ARP complex with partial Glu-Phe ARPs inclusion. The proposed structure of the CD-ARP complex indicated that Glu-Phe ARPs were deeply inserted into the β-CD cavity from the broad side [27].

2D NMR provides the most direct evidence for observing the spatial proximity between the host and guest atoms following intermolecular dipole cross-correlation. In the ROESY spectrum, two protons that are close in space can produce a nuclear Overhauser effect (NOE) cross-correlation. The ROESY spectrum (Figure 3c) of the CD-ARP complex showed an obvious correlation between the H-b (f) of Glu-Phe ARPs and H-5 of β-CD, indicating that the phenyl ring of Glu-Phe ARPs was incorporated into the β-CD cavity (Figure 3d). These observations were not surprising because the most likely mode of binding in the β-CD complex was the partial incorporation of the hydrophobic group of the guest into the cavity. Encapsulation of the hydrophobic side chain that contributed to the bitter taste of Glu-Phe ARPs into the cage-like structure could reduce bitter chain exposure to taste buds [28]. Thus, it can be assumed that the CD-ARP complex cannot interact with the bitter taste receptor to induce a bitter taste reaction.

### 3.4. Analysis of Computational Molecular Docking with hT2R1

The synergistic effects of Glu-Phe ARPs, Phe, and CD-ARP complex on bitter receptor hT2R1 were studied using computational molecular modelling and molecular docking to reveal the taste mechanism of bitter ingredients. Compared to other bitter receptors (e.g., hT2R4 and hT2R14), hT2R1, selected as an ideal candidate, could interact with Phe at a similar activity level to that of the reference agonist menthol (300 μM) [29]. The model verification results indicated that the constructed hT2R1 receptor model containing 299 amino acid residues with 98.66% of those in the allowed region and 1.34% in the disallowed region (Figure 4a), was reasonable on the basis of dihedral distribution and steric clashes. In addition, the active sphere of hT2R1 for ligand binding was located close to the extracellular surface (Figure 4b), indicating that the less conserved active sphere of hT2R1 could form different binding motifs with numerous and structurally diverse bitter compounds [30]. 

We searched for the best docking position with minimum docking energy and interaction energy (IE). The interaction sites of hT2R1 with the bitter compounds were displayed as 2D diagrams (Figure 4c,d). The results indicated that the bitter compounds Glu-Phe ARPs and Phe could be docked on hT2R1 at the same interaction sites, such as Ser 271, Gly 272, and Lys 283, indicating that the active region of hT2R1 that interacted with Glu-Phe ARPs was close to that of Phe. Moreover, other interaction sites of hT2R1 (such as Glu 182 and Ala 288) with Glu-Phe ARPs were different from those of Phe. These results could be related to the glycosyl moiety of Glu-Phe ARPs with respect to Phe, which induced the different orientations of Glu-Phe ARPs interacting with hT2R1 compared to those of Phe. Unexpectedly, the IE of Glu-Phe ARPs (−29.2567 kcal/mol) with hT2R1 was close to that of Phe (−28.7045 kcal/mol). This result indicated that Glu-Phe ARPs were active for the hT2R1 bitter receptor due to the bitter activity of Phe on hT2R1. The results confirmed the bitter activity of Glu-Phe ARPs at the molecular level and the bitter taste was similar to that of Phe. In contrast, the encapsulated Glu-Phe ARPs with β-CD could not successfully bind to hT2R1 with the same docking method, indicating that the CD-ARP complex could be a potential compound with a non-bitterness taste.

### 3.5. Thermal Degradation Kinetics

The thermal degradation of Glu-Phe ARPs, as an MR intermediate, is particularly important for the formation of the flavor and color of heated food. Studying the thermal degradation kinetics of Glu-Phe ARPs and CD-ARP complexes can improve the heat treatment conditions to maximize the rate of production of desired flavor compounds and inhibit the rate of production of off-flavors or undesirable products [31].

As shown in Figure 5, the retention (%) of Glu-Phe ARPs and CD-ARP complexes decreased with time at different temperatures (80, 90, and 100 °C). There was a linear relationship (R^2^ > 0.9) between ln (C/C_0_) and time for Glu-Phe ARPs and the CD-ARP complex, clearly indicating that Glu-Phe ARPs degradation followed first-order reaction kinetics [32]. The k values of Glu-Phe ARPs increased in a temperature-dependent manner. And the k value of Glu-Phe ARPs at 100 °C (0.900 h^−1^) was 7.5 times higher than that at 80 °C (0.120 h^−1^), and t_1/2_ decreased from 5.796 h to 0.771 h. The k values of the CD-ARP complex were significantly lower than those of Glu-Phe ARPs, but increased slowly with increasing temperature. The k value at 100 °C (0.044 h^−1^) was 1.6 times higher than that at 80 °C (0.028 h^−1^), and t_1/2_ decreased from 24.407 h to 15.898 h. The results showed that the CD-ARP complex exhibited excellent thermal stability compared to the free Glu-Phe ARPs.

### 3.6. Color Values of Cookie Samples with the CD-ARP Complex

The brown polymers produced by the MR, called melanoidins, are largely responsible for the color development of heat-treated foods. However, excessive melanoidins can darken the color of food and reduce product acceptance [33]. To evaluate the effect of the addition of Glu-Phe ARPs, CD-ARP complex, and Glu + Phe on the color of the baked biscuit, several scalar parameters (*L**, *a**, and *b**) were measured (Table 2). Compared with the blank group, adding Glu-Phe ARPs to the cookie samples decreased the *L** value (brightness), but increased the *a** value (redness) and *b** value (yellowness). This indicated that the color of the cookie was affected by the addition of Glu-Phe ARPs. However, the CD-ARP complex groups had significantly higher *L** values and lower *a** values compared to the Glu-Phe ARPs groups, and were closer to the blank group. In addition, the calculated Δ*E* also indicated significant color differences between the groups with the addition of Glu-Phe ARPs and the CD-ARP complex. The encapsulation of Glu-Phe ARPs with β-CD could effectively inhibit the Maillard browning from Glu-Phe ARPs. Thus, the application of the CD-ARP complex could improve the sensory quality of cookies.

### 3.7. Flavor Identification of Cookie Samples with the CD-ARP Complex

Benzaldehyde and phenylacetaldehyde are the characteristic flavor compounds produced by the MR of Phe, which can give cookies a floral-fruity fragrance [34]. GC-MS was used to identify and quantitate benzaldehyde and phenylacetaldehyde from different baked cookie samples. Figure 6a showed the extracted ion chromatograms of *m*/*z* 106 and 91 for benzaldehyde and phenylacetaldehyde in the cookies with CD-ARP complex (0.5 g). These compounds were not detected in the control group. As shown in Figure 6b,c, the target flavor compounds were detected with the addition of Glu-Phe ARPs, CD-ARP complex, and Glu + Phe, and the amount of detected flavor compounds increased with the amount added. When Glu-Phe ARPs (0.5 g) were added, the detected concentrations of benzaldehyde and phenylacetaldehyde were 0.83 ± 0.04 µg/g and 4.77 ± 0.30 µg/g, respectively. The concentrations of benzaldehyde and phenylacetaldehyde (0.23 ± 0.06 µg/g and 0.54 ± 0.04 µg/g) with Glu + Phe as flavor precursors were lower than those in cookies with Glu-Phe ARPs, which directly proved that the reactivity of Glu + Phe in MR was inferior to that of Glu-Phe ARPs [35]. In addition, when the CD-ARP complex containing 0.5 g of Glu-Phe ARPs was added, the detected concentrations of benzaldehyde and phenylacetaldehyde were 0.88 ± 0.17 µg/g and 4.71 ± 1.20 µg/g, respectively, which were similar to those observed with the addition of Glu-Phe ARPs, indicating that the CD-ARP complex had good flavor release properties during heat treatment.

## 4. Conclusions

In this study, the CD-ARP complex could be applied as an efficient flavor enhancer with low brown intensity in a thermally treated food system compared to direct Glu-Phe ARPs application. The elucidated chemical structure of the CD-ARP complex indicated that the stoichiometric ratio of Glu-Phe ARPs-β-CD complex was 1:1, and the benzene ring moiety of Glu-Phe ARPs was inserted in the interior cavity of β-CD. Using a docking analysis of homology modelling of hT2R1, the fact that Glu-Phe ARPs could effectively bind to the bitter receptor hT2R1 rather than the CD-ARP complex confirmed that the bitter activity of Glu-Phe ARPs could be reduced by β-CD encapsulation. Thus, β-CD encapsulation of Glu-Phe ARPs slowed down the formation of brown compounds but did not inhibit flavor compound formation in the food system. Further studies are necessary to investigate the mechanism of reducing brown intensity by embedding ARPs with β-CD and provide additional opportunities to expand the utilization of ARPs.

## Figures and Tables

**Figure 1 foods-11-01309-f001:**
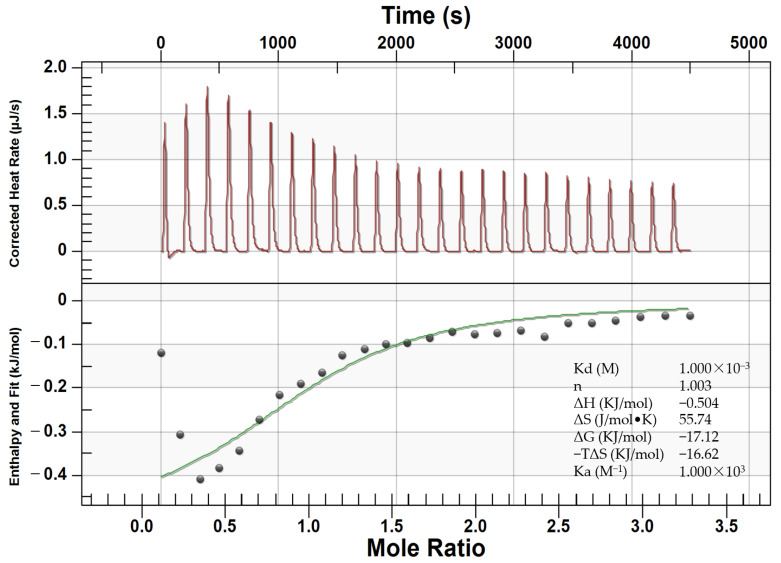
Thermodynamics of binding interactions between Glu-Phe ARPs and β-CD as measured using ITC. The upper panel displays a representative calorimetric titration curve. The lower panel exhibits a fitted curve based on an independent binding model.

**Figure 2 foods-11-01309-f002:**
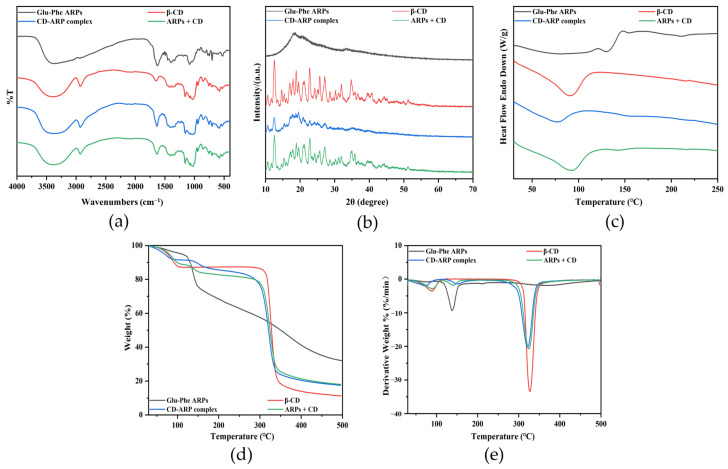
(**a**) FT-IR spectra, (**b**) XRD patterns, (**c**) DSC curves, (**d**) TG and (**e**) DTG curves of Glu-Phe ARPs, β-CD, CD-ARP complex and ARPs + CD.

**Figure 3 foods-11-01309-f003:**
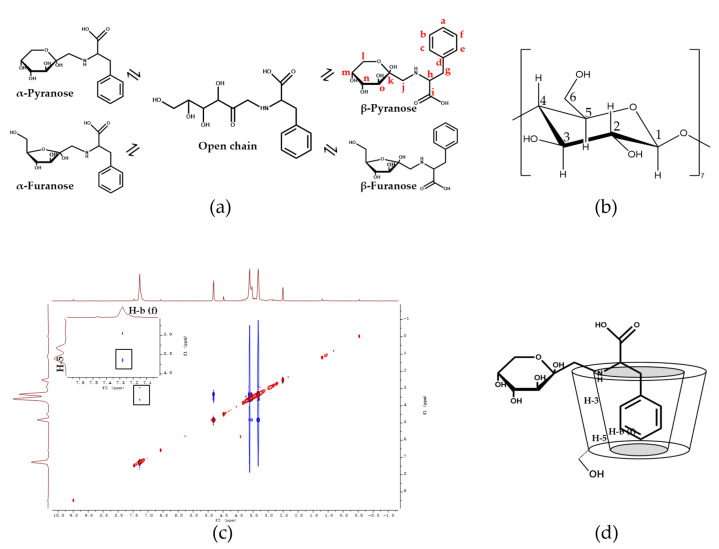
(**a**) Tautomeric forms of the Glu-Phe ARPs, (**b**) structure of β-CD showing the numbering of the β-glucose monomer, (**c**) ROESY spectrum of the CD-ARP complex and (**d**) possible CD-ARP complex mode.

**Figure 4 foods-11-01309-f004:**
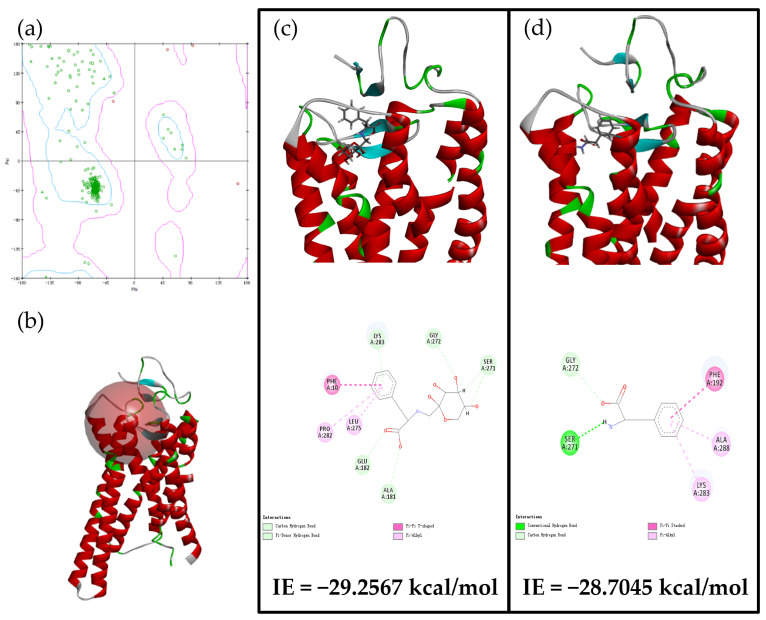
(**a**) The Ramachandran plot of the hT2R1 model, (**b**) the active sphere of the hT2R1 model, (**c**) molecular docking results and IE of Glu-Phe ARPs and (**d**) Phe in the hT2R1 receptor active site.

**Figure 5 foods-11-01309-f005:**
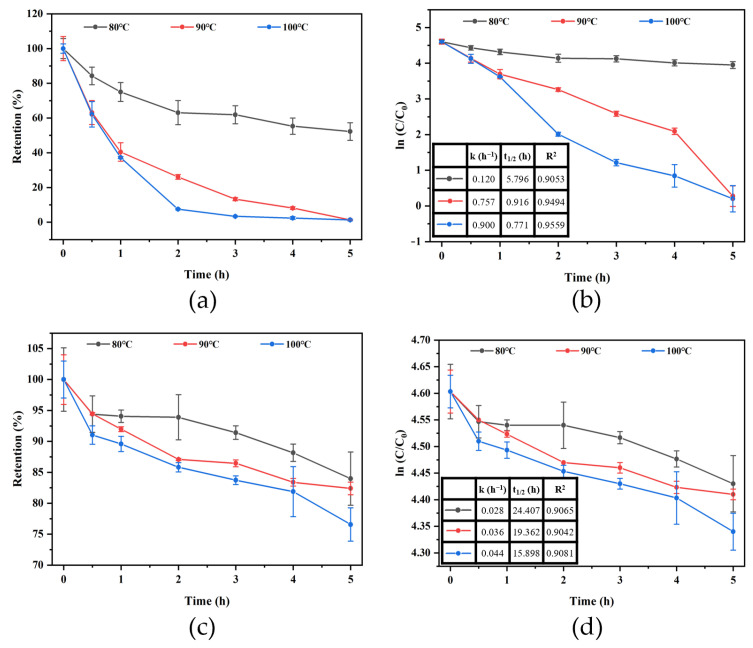
Plots of retention and ln (C/C_0_) vs. heating time with Glu-Phe ARPs (**a**,**b**) and the CD-ARP complex (**c**,**d**) at different temperatures.

**Figure 6 foods-11-01309-f006:**
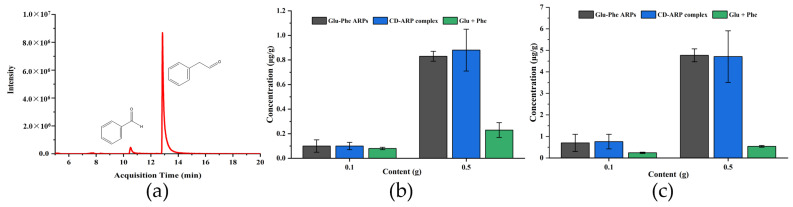
(**a**) Extracted ion chromatogram of cookies with CD-ARP complex (0.5 g). The benzaldehyde (**b**) and phenylacetaldehyde (**c**) concentrations of cookies with Glu-Phe ARPs, CD-ARP complex and Glu + Phe. Content is the mass of the added or theoretically generated Glu-Phe ARPs.

**Table 1 foods-11-01309-t001:** Variation of proton chemical shift (Δδ/ppm) of β-CD before and after forming complex with Glu-Phe ARPs.

^1^ H Assignment	β-CD	CD-ARP Complex	
	δ (ppm)	δ (ppm)	Δδ ^1^ (ppm)
H-1	4.820	4.832	0.012
H-2	3.550	3.519	−0.031
H-3	3.652	3.804	0.152
H-4	3.330	3.311	−0.019
H-5	3.631	3.604	−0.027
H-6	3.631	3.600	−0.031

^1^ A positive sign of Δδ ppm shows a downfield displacement and a negative sign an upfield displacement (Δδ = δ complex − δ free).

**Table 2 foods-11-01309-t002:** Effect of Glu-Phe ARPs and CD-ARP complex addition on the color properties of cookies.

Sample	Content ^1^/g	*L**	*a**	*b**	Δ*E*	Cookie Color
Blank	0	76.45 ± 0.41 ^a^	1.87 ± 0.08 ^k^	26.28 ± 0.41 ^h^	—	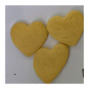
Glu-PheARPs	0.1	69.98 ± 0.50 ^c^	8.92 ± 0.19 ^g^	37.03 ± 0.47 ^c,d^	—	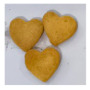
0.2	68.12 ± 0.08 ^d^	10.29 ± 0.18 ^f^	38.46 ± 0.16 ^b^	—	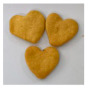
0.3	63.22 ± 1.68 ^e^	12.02 ± 0.12 ^d^	37.35 ± 0.71 ^c^	—	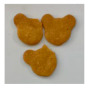
0.4	55.32 ± 0.81 ^g^	14.34 ± 0.27 ^b^	35.67 ± 0.75 ^e^	—	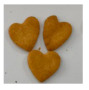
0.5	51.95 ± 1.17 ^f^	17.46 ± 0.30 ^a^	35.81 ± 0.56 ^e^	—	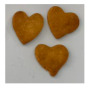
CD-ARPcomplex	0.1	69.22 ± 0.65 ^c,d^	4.11 ± 0.06 ^j^	32.16 ± 0.42 ^f^	6.91 ± 0.29 ^d^	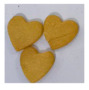
0.2	73.42 ± 0.38 ^b^	6.30 ± 0.07 ^i^	35.43 ± 0.59 ^e^	6.84 ± 0.47 ^d^	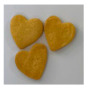
0.3	69.55 ± 1.25 ^c,d^	7.53 ± 0.16 ^h^	36.10 ± 0.59 ^d,e^	7.90 ± 0.96 ^c^	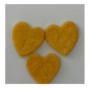
0.4	68.75 ± 0.41 ^c,d^	11.49 ± 0.35 ^e^	40.53 ± 1.19 ^a^	17.76 ± 0.29 ^b^	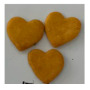
0.5	62.58 ± 0.45 ^e^	13.29 ± 0.18 ^c^	37.82 ± 0.68 ^b,c^	8.63 ± 0.63 ^c^	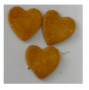
Glu + Phe	0.5	74.74 ± 0.28 ^b^	2.07 ± 0.02 ^k^	27.67 ± 0.05 ^g^	26.09 ± 0.20 ^a^	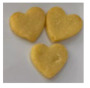

—, not calculated. Values are mean ± standard deviation of three separate determinations (*n* = 3). Means followed by a–k letters in the same column for each component are significantly different (*p* < 0.05). ^1^ Content is the mass of the added or theoretically generated Glu-Phe ARPs.

## Data Availability

The datasets generated for this study are available on request to the corresponding author.

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
