# Peer review of "A Debittered Complex of Glucose-Phenylalanine Amadori Rearrangement Products with β-Cyclodextrin: Structure, Molecular Docking and Thermal Degradation Kinetic Study"

_foods, 2022, doi:10.3390/foods11091309_

Round 1

Reviewer 1 Report

The authors provide extensive information regarding ARP/cyclodextrin complex formation and its thermodynamic and spectroscopic properties and its application and its role in color formation in baked cookies. However two issues I want to point out regarding the conclusions reached.

(1) β-cyclodextrin complexes are known to dissociate thermally and in solution. Although the authors measured the rate of its decomposition but this data does not provide information on the dissociation of the complex. How much of the complex is releasing free ARP before baking for example once it is added to the mixture. How stable is the complex over time in terms of dissociation and release of free ARP at specific temperatures and in specific food matrices? I think this is an important information should be included if available.

(2) The authors rely only on molecular docking data to conclude that the bitter taste of the ARP will be reduced. Is this reliable? Have they performed actual taste testing of the cookies to confirm that complexed ARP had reduced bitter taste relative to cookies baked with free ARP?

Author Response

Comment#1: β-cyclodextrin complexes are known to dissociate thermally and in solution. Although the authors measured the rate of its decomposition but this data does not provide information on the dissociation of the complex. How much of the complex is releasing free ARP before baking for example once it is added to the mixture. How stable is the complex over time in terms of dissociation and release of free ARP at specific temperatures and in specific food matrices? I think this is an important information should be included if available.

Answer#1: Thank you very much for your suggestion. First, we determined the inclusion efficiency (IE) of the CD-ARP complex to be 86.79±0.23 (%). The inclusion effect is good. The method was as follows: A total of 0.1 g of CD-ARP complex will be extracted with 5 mL of anhydrous ethanol for free Glu-Phe ARPs. The supernatant obtained was used for HPLC- DAD analysis. IE was calculated using equation (1), where A is initial Glu-Phe ARPs mass to be encapsulated, and B is the mass of free Glu-Phe ARPs.

IE (%) = (A-B)/B*100 (1)

Second, the prepared Glu-Phe ARPs can generate violet fragrance when heated[1]. Generally, the application field of CD-ARP complex is bakery products. Bakery products consist essentially of the solid matrix, so the CD-ARP complex does not dissociate easily. In addition, our manuscript also provided experiment related to the application of bakery products. The results of color experiment (Section 3.6.) showed that cookies added with Glu-Phe ARPs and CD-ARP complex had obvious differences in color after baking, such as the calculation results of color difference (ΔE). The color of the CD-ARP complex groups were lower than that of the Glu-Phe ARPs groups and closed to that of the blank group. According to the above results, CD-ARP complex can’t be dissociated or ignored during application, and the application effect is not affected.

[1] Cui, H.; Hayat, K.; Zhang, X. Antioxidant activity in vitro of N-(1-deoxy-alpha-d-xylulos-1-yl)-phenylalanine: comparison among Maillard reaction intermediate, end-products and xylose-phenylalanine. J. Food Sci. 2019, 84, 1060-1067, doi:10.1111/1750-3841.14579.

Comment#2: The authors rely only on molecular docking data to conclude that the bitter taste of the ARP will be reduced. Is this reliable? Have they performed actual taste testing of the cookies to confirm that complexed ARP had reduced bitter taste relative to cookies baked with free ARP?

Answer#2: Thanks for your suggestion. After the synthesis of Glu-Phe ARPs and CD-ARP complex, we conducted the evaluation experiment on the effect of β-cyclodextrin on debittering Glu-Phe ARPs. The specific test procedures were in reference to the method of Marzouk et al.[1] and modified accordingly. Each group member tasted 0.05 g Glu-Phe ARPs, CD-ARP complex and physical mixture (ARPs + CD), and was given their sensory score for the degree of bitterness within 10 seconds. The response was evaluated on a scale from 0 to 4 (0: no bitter, 1: tasteless, 2: slightly bitter, 3: bitter, and 4: very bitter). L-phenylalanine was selected as the bitter reference substance and its bitter grade was specified as 3. The results are presented in the Table 1. The results showed that the bitterness of Glu-Phe ARPs embedded with β-cyclodextrin was significantly reduced. At the same time, the bitter taste of cookies was analyzed in the same way. The results showed that compared with the addition of free Glu-Phe ARPs group, the bitter taste of cookies with CD-ARP complex was reduced to the completely acceptable degree.

Table 1 Evaluation of debittering effect of the CD-ARP complex based on panel test

No of volunteers

1

2

3

4

5

6

Sample

Taste score after 10 s

   mean ± SD

Glu-Phe ARPs

4

3

3

3

4

3

  3.33 ± 0.52a

CD-ARP complex

0

0

0

1

0

1

  0.33±0.52c

ARPs + CD

3

2

2

2

2

2

  2.17±0.41b

Values in the row with the same letter in superscript are not significant different from each other at P<0.05.

Reviewer 2 Report

This manuscript discusses the application of ß-cyclodextrin for masking the bitter taste of aldehydes as off-flavours. Considering the recent interest in the application of cyclodextrins for encapsulation and delivery of flavours in foods, this article is timely and might be of interest to food researchers and scientists. The performed experimental tests are discussed well and the required details are provided. However, the article can be strengthened by adding more information in some sections.

I have the following comments and questions.

Comment#1

  • In the keywords, pay attention to the capitalization of the words

Comment#2

  • On Page 1, line 41 the authors stated that in their previous test Glu-Phe ARPs could introduce a bitter taste. Provide a reference here for the previous observations.

Comment#3

  • On page 2, line 45 it is stated that beta-cyclodextrin is commonly used for encapsulating bitter molecules. Please comment on why other natural forms of cyclodextrins such as alpha or gamma cyclodextrins are not used.

Comment#4

  • In equation 1, page 3 the “t” for expressing time after “k” parameter is missing. Please correct. The equation should be presented as:

Ln(C/C0)=-kt

Comment#5

  • In line 156, page 4, provide the complete word “dimethylsulfoxide” for DMSO-d6. The abbreviated term is used but not explained in the text.

Comment#6

  • On page 5, line 208 it is explained that the value of DeltaH was observed as small and slightly negative and elevation in temperature resulted in dissociation of the binding. indicate how much was the temperature increase that resulted in the observed dissociation.

Comment#7

  • On page 6, line 229, provide a reference for the mentioned wave numbers for Glu-Phe ARPs.

Comment#8

  • On page 6, line 229, please comment if the peak observed related to the aromatic ring of Glu-Phe is present in the ARPs-CD complex and if the ring is included inside the cavity and similar results as NMR studies can be obtained.(Magnification of the spectra between 1500-2000 wavenumber)

Comment#9

  • On page 11, line 380, in the sentence “were higher than those of the control samples, but no significant differences were observed.” It is mentioned that the difference was high but not significant for the values of b*. The Delta values should be calculated for (sample-control) and based on the values of Delta b*, authors can comment on the observed difference between samples.

Similarly, the values of Delta L*, and Delta a* should be calculated between samples and control and based on the values of Deltas (+/-) authors can comment on the differences between samples in terms of lightness/darkness, yellowness/blueness, and redness/greenness.

Also, the value of Delta E that represents the colour difference between samples should be calculated from the delta values from the formula below:

(Delta E)=SQRT [ (Delta L*)^2+(Delta a*)^2+(Delta b*)^2 ]

And based on the acceptability of the obtained ranges of Delta E (commonly in the excellent range between 0-4) the authors can comment on how encapsulation with cyclodextrins and different concentrations of  Glu-Phe-ARPs can influence the colour change.

Values of Delta L*, Delta a*, Delta b*, and Delta E should be added to Table 2 for a better comparison between samples as these three parameters together define the colour.

Comment#10

  • In the conclusions section, discuss more on the future direction of the research in this area. E.g. what is suggested for inhibiting the formation of brown colour in samples and reducing the values of Delta E.

Author Response

Comment#1: In the keywords, pay attention to the capitalization of the words

Answer#1: Thank you very much for your suggestion. We have checked the capitalization of the words in the keywords. Among them, “Amadori” is a proper noun, and it is often used in the form of uppercase.

Comment#2: On Page 1, line 41 the authors stated that in their previous test Glu-Phe ARPs could introduce a bitter taste. Provide a reference here for the previous observations.

Answer#2: Thanks for your suggestion. After the synthesis of Glu-Phe ARPs, we carried out the taste test on it. The specific test procedures were in reference to the method of Marzouk et al.[1] and modified accordingly. Each group member tasted 0.05 g Glu-Phe ARPs and was given their sensory score for the degree of bitterness within 10 seconds. The response was evaluated on a scale from 0 to 4 (0: no bitter, 1: tasteless, 2: slightly bitter, 3: bitter, and 4: very bitter). L-phenylalanine was selected as the bitter reference substance and its bitter grade was specified as 3. The results are presented in the Table 1.

Table 1 Evaluation of the bitter taste of Glu-Phe ARPs

No of volunteers

1

2

3

4

5

6

Sample

Taste score after 10 s

   mean ± SD

Glu-Phe ARPs

4

3

3

3

4

3

  3.33 ± 0.52

Comment#3: On page 2, line 45 it is stated that beta-cyclodextrin is commonly used for encapsulating bitter molecules. Please comment on why other natural forms of cyclodextrins such as alpha or gamma cyclodextrins are not used.

Answer#3: Thanks for your suggestion. Firstly, α-cyclodextrin has a small internal cavity size and can only encapsulate small molecules. Secondly, despite the large internal cavity size of γ-cyclodextrin, it is costly and cannot be produced in large quantities. Therefore, their application are limited. But the β-cyclodextrin is one of the most widely used cyclodextrin products in industry due to its moderate inner cavity size, good selectivity in binding to various guest molecules, non-toxicity and low cost[1, 2].

[1] Abarca, R.L.; Rodriguez, F.J.; Guarda, A.; Galotto, M.J.; Bruna, J.E. Characterization of beta-cyclodextrin inclusion complexes containing an essential oil component. Food Chem. 2016, 196, 968-975, doi:10.1016/j.foodchem.2015.10.023.

[2] Astray, G.; Mejuto, J.C.; Morales, J.; Rial-Otero, R.; Simal-Gándara, J. Factors controlling flavors binding constants to cyclodextrins and their applications in foods. Food Res. Int. 2010, 43, 1212-1218, doi:10.1016/j.foodres.2010.02.017.

Comment#4: In equation 1, page 3 the “t” for expressing time after “k” parameter is missing. Please correct. The equation should be presented as: Ln(C/C0)=-kt

Answer#4: Thank you for pointing out the problem. We have revised the formula in the manuscript. The relative details have been modified in the P.3, L.105.

Comment#5: In line 156, page 4, provide the complete word “dimethylsulfoxide” for DMSO-d6. The abbreviated term is used but not explained in the text.

Answer#5: Thank you for your advice. We have provided the complete word “dimethylsulfoxide” for DMSO-d6 in the manuscript. The relative details have been modified in the P.4, L.155.

Comment#6: On page 5, line 208 it is explained that the value of DeltaH was observed as small and slightly negative and elevation in temperature resulted in dissociation of the binding. indicate how much was the temperature increase that resulted in the observed dissociation.

Answer#6: Thank you for your advice. We would like to describe to you that the experimental results show a value of -0.504 kJ mol-1 for ΔH, indicating that the binding reaction is exothermic. And the exothermic reactions have the characteristics of difficult combination when the temperature is increased, i.e., easy dissociation.

Comment#7: On page 6, line 229, provide a reference for the mentioned wave numbers for Glu-Phe ARPs.

Answer#7: Thank you for your advice on our manuscript. The reference has been added in the P.14, L.485-486.

Comment#8: On page 6, line 229, please comment if the peak observed related to the aromatic ring of Glu-Phe is present in the ARPs-CD complex and if the ring is included inside the cavity and similar results as NMR studies can be obtained.(Magnification of the spectra between 1500-2000 wavenumber)

Answer#8: Thank you for your suggestion on our manuscript. We reanalyzed the FT-IR data. Since that prepared CD-ARP complex contained a small amount of free Glu-Phe ARPs, the characteristic peak of the aromatic ring can be detected by FT-IR. Furthermore, the C=C stretching vibration peak in the benzene ring conjugated system of Glu-Phe ARPs at 1627 cm-1 shifted to 1634 cm-1 after the formation of the CD-ARP complex. This result indicated that β-CD interacted with Glu-Phe ARPs to form CD-ARP complexes.

Comment#9: On page 11, line 380, in the sentence “were higher than those of the control samples, but no significant differences were observed.” It is mentioned that the difference was high but not significant for the values of b*. The Delta values should be calculated for (sample-control) and based on the values of Delta b*, authors can comment on the observed difference between samples.

Similarly, the values of Delta L*, and Delta a* should be calculated between samples and control and based on the values of Deltas (+/-) authors can comment on the differences between samples in terms of lightness/darkness, yellowness/blueness, and redness/greenness.

Also, the value of Delta E that represents the colour difference between samples should be calculated from the delta values from the formula below:

(Delta E)=SQRT [ (Delta L*)^2+(Delta a*)^2+(Delta b*)^2 ]

And based on the acceptability of the obtained ranges of Delta E (commonly in the excellent range between 0-4) the authors can comment on how encapsulation with cyclodextrins and different concentrations of Glu-Phe-ARPs can influence the colour change.

Values of Delta L*, Delta a*, Delta b*, and Delta E should be added to Table 2 for a better comparison between samples as these three parameters together define the color.

Answer#9: Thank you for your suggestion. According to your suggestion, corresponding descriptions have been changed in the P.11-12 and Table 2.

Comment#10: In the conclusions section, discuss more on the future direction of the research in this area. E.g. what is suggested for inhibiting the formation of brown colour in samples and reducing the values of Delta E.

Answer#10: Thank you very much for your suggestion. According to your suggestion, relative description has been added in the P.13, L.425-427.
